# Generalized Solution of Inverse Problem for Ising Connection Matrix on *d*-Dimensional Hypercubic Lattice

**DOI:** 10.3390/e24101424

**Published:** 2022-10-06

**Authors:** Boris Kryzhanovsky, Leonid Litinskii

**Affiliations:** Center of Optical Neural Technologies, Scientific Research Institute for System Analysis, Russian Academy of Sciences, Nakhimov Ave, 36-1, 117218 Moscow, Russia

**Keywords:** Ising connection matrix, eigenvalues, Kronecker product, inverse problem

## Abstract

We analyze a connection matrix of a d-dimensional Ising system and solve the inverse problem, restoring the constants of interaction between spins, based on the known spectrum of its eigenvalues. When the boundary conditions are periodic, we can account for interactions between spins that are arbitrarily far. In the case of the free boundary conditions, we have to restrict ourselves with interactions between the given spin and the spins of the first d coordination spheres.

## 1. Introduction

Around one hundred years ago, a group of scientists made the Ising model one of the subjects of their research. They obtained exact solutions [1] for some realistic spin systems; when the problem could not be solved exactly, methods of computer simulation were developed. The Ising model is applicable far beyond the investigations of the magnetic properties of different materials. In the review [2], we can find the formulation of the combinatorial optimization problems, in terms of the Ising model. The Hopfield model of the associative neural network [3] is a neural analog of the Ising model with long-range interaction. The Ising model is helpful in the problems of the socio- and econophysics [4] and in many other fields of research.

Some time ago, we developed an m-vicinity method for calculation of the partition function [5], which showed itself rather effective. The necessity to calculate exact expressions for the eigenvalues of the Ising connection matrix arose from this research. The connection matrix J defines the energy $E(s)\sim -sJs^{+}$ of the state $s=\left(s_{1},s_{2},\ldots ,s_{N}\right),\;s_{i}=\pm 1$. Consequently, the density of the energy distribution and the free energy, as well as all the macroscopic characteristics of the system depend on the connection matrix J, while the set of the eigenvalues characterize the connection matrix exhaustively. (Some details we present in the end of Section 5.)

The authors of paper [6] were the first who obtained the eigenvalues of the connection matrices of the d-dimensional Ising models for d=1,2,3. They assumed the periodic boundary conditions and accounted for the interaction with the nearest neighbors.

In paper [7], we succeeded in calculating exact expressions for the eigenvalues, accounting for the interactions with the nearest and the next-nearest neighbors. We also discussed the periodic boundary conditions. From our results, it became evident that independently of the lattice dimension d, the result was defined by the one-dimensional connection matrices J(k), which accounted for the interactions with the k-th neighbors only.

In paper [8], we examined the lattice dimensions d=1,2,3 and accounted for the interactions with the nearest, the next-nearest, and the next-next-nearest neighbors. In addition, we found out that there was a principal difference between the periodic and free boundary conditions. In the first case, all the matrices J(k) commuted. Consequently, these matrices had the same set of the eigenvectors, and this circumstance crucially simplified the problem. On the contrary, when the boundary conditions were free, each matrix J(k) had its own set of the eigenvectors. These matrices did not commute, and this obstacle restricted the applicability of our approach.

In paper [9], we assumed the periodic boundary conditions and examined the d-dimensional Ising system. We showed that the eigenvectors of its connection matrix were the Kronecker products of the well-known eigenvectors of the one-dimensional model and obtained the expressions for the corresponding eigenvalues in the form of the polynomials of degree d in powers of the eigenvalues of the one-dimensional system. The constants of interaction between spins played the role of the coefficients of these polynomials. We found that the account of long-range interactions did not lift the degeneracy of the discrete spectrum.

Finally, in paper [10], we solved the inverse problem, restoring the interaction constants from the known spectrum of the connection matrix, when d was equal to 1, 2, and 3. In the present paper, we generalize these results in the case of the hypercubic lattice of an arbitrary dimension d>3.

In Section 4, we present new results obtained in this paper. They are:(a)The general formulas allow us to determine the interaction constants when we know the spectrum of the eigenvalues of the connection matrix of some spin system with an arbitrary long-range interaction on a hypercube lattice of an arbitrary dimension d (see Equation (11) and below);(b)An exhaustive analysis of the application of our approach to the spin systems with free boundary conditions (Section 4.2). This is an important result because of the difficulties arising when examining such systems.

As we have mentioned above, to construct the connection matrix of the multidimensional model and to write down its eigenvectors and eigenvalues, it is sufficient to know the one-dimensional matrices J(k), their eigenvectors, and eigenvalues. Consequently, in Section 2 we discuss briefly the main results for the one-dimensional model with periodic boundary conditions. In Section 3, we generalize these results to the case of the two-dimensional model, with the aid of the technique of the Kronecker products. In Section 4, we derive the final formulas for the connection matrix of the Ising model on the hypercubic lattice of an arbitrary dimension d and discuss the case of the free boundary conditions. The conclusions are in Section 5.

## 2. One-Dimensional Ising Model

Let us examine a one-dimensional chain consisting of L spins and suppose the periodic boundary conditions. To be definite, we assume that L is odd: L=2l+1. This assumption is not important, since the case of an even L can be examined in the same way.

When L is odd, each spin has two nearest neighbors, two next-nearest neighbors, two next-next-nearest neighbors, …, and two l-th neighbors. By J(k), we denote an (L×L)- matrix where we account for interactions with the k-th neighbors only. It is easy to see that J(k) is a symmetric matrix with the ones on the k-th and the (L−k)-th overdiagonals and underdiagonals. Other elements of these matrices are equal to zero.

We can write the elements of the matrix J(k) as
(1)$$J_{ij}(k)=\delta_{i_{1},j}+\delta_{i_{2},j},$$
where
$$xi_{1}=\{\begin{array}{c} i+k,\;\;\;if\;i+k\leq L\\ i+k-L,\;if\;i+k>L \end{array},\;\;\;\;\;i_{2}=\{\begin{array}{c} i-k,\;\;\;if\;i-k>0\\ i-k+L,\;if\;i-k\leq 0 \end{array}.$$

In total, we have l matrices J(k). Each matrix J(k) is a circulant matrix that is a square matrix, where each subsequent row is cyclic-shifted by one element to the right relative to the previous row. Consequently, all the matrices J(k) have the same set of the eigenvectors ${\left\{f_{\alpha }\right\}}_{\alpha =1}^{L}$ (see [11,12]). At the same time each matrix, J(k), has its own set of the eigenvalues ${\left\{\lambda_{i}(k)\right\}}_{i=1}^{L}$. For our purposes, it is convenient to introduce an L-dimensional vector λ(k) whose components are the eigenvalues $\lambda_{i}(k)$. The expressions
(2)$$f_{j}(\alpha )=\frac{1}{\sqrt{L}}\times \{\begin{array}{c} 1\;,\\ \;\sqrt{2}\cdot \cos\;\left[(j-1)\;\varphi_{\alpha }\right]\;,\;\;\;\;\\ \sqrt{2}\cdot \sin\left[(j-1)\varphi_{\alpha }\right],\;\;\; \end{array}\begin{array}{c} \alpha =1\\ 2\leq \alpha \leq l+1\\ l+2\leq \alpha \leq L \end{array}$$
(3)$$\lambda_{\alpha }(k)=2\cdot \cos(k\varphi_{\alpha }),$$
define the vectors $f_{\alpha }=\left(f_{1}(\alpha ),f_{2}(\alpha ),\ldots ,f_{L}(\alpha )\right)$ and $\lambda (k)=\left(\lambda_{1}(k),\lambda_{2}(k),\ldots ,\lambda_{L}(k)\right)$, where $\varphi_{\alpha }=2\pi (\alpha -1)/L$, α=1,2,…,L, k=1,2,…,l, and j=1,2,…,L (see [7]).

The first eigenvalue of J(k) (1) is equal to two and its other eigenvalues are two times degenerate: $\lambda_{\beta }(k)=\lambda_{L+2-\beta }(k)$; β=2,3,…,l+1. It is easy to see that the vectors λ(k) are mutually orthogonal because each vector λ(k) is collinear to the eigenvector $f_{k+1}$:(4)$$\lambda (k)\;\lambda^{+}(r)=2L\cdot \delta_{kr},\;\lambda (k)\;=\sqrt{2L}\cdot f_{k+1}.$$

Consequently, in the one-dimensional case we have l matrices J(k) and the corresponding l “eigenvalue vectors” λ(k) (k=1,2,…,l) and L eigenvectors $f_{\alpha }$, α=1,2,…,L. These characteristics are sufficient to analyze the Ising model on an arbitrary d-dimensional lattice.

We can use the matrices ${\left\{J(k)\right\}}_{k=1}^{l}$ to write down the connection matrix of the one-dimensional chain in the general form:(5)$$U_{1}=\sum_{k=1}^{l}w(k)J(k).$$In Equation (5), w(1) is the constant of interaction with the nearest neighbors, w(2) is the constant of interaction with the next-nearest neighbors, and so on. Finally, w(l) is the constant of interaction with the l-th neighbors.

It is evident that the set of the eigenvectors of the matrix, $U_{1}$, consist of the eigenvectors ${\left\{f_{\alpha }\right\}}_{\alpha =1}^{L}$, defined in Equation (2). The eigenvalues ${\left\{\mu_{1}(\alpha )\right\}}_{\alpha =1}^{L}$ of this matrix can be written as a combination of the vectors λ(k) (see Equation (5)):(6)$$\mu_{1}=\sum_{k=1}^{l}w(k)\;\lambda (k).$$

The equality (6) allows us to solve the inverse problem for the matrix $U_{1}$. Indeed, suppose we know the spectrum of the eigenvalues of the matrix $U_{1}$. That means that we know all the components of the vector $\mu_{1}$ in the left-hand side of Equation (6). Can we then restore the matrix $U_{1}$, that is to determine the interaction constants ${\left\{w(k)\right\}}_{k=1}^{l}$ in Equation (5)? The answer is very simple. When multiplying both sides of the equality (6) by the column vector $\lambda^{+}(k)$ and taking into account the orthogonality condition (4), we obtain
$$w(k)=\frac{\mu_{1}\lambda^{+}(k)}{{\left\|\lambda (k)\right\|}^{2}}=\frac{\mu_{1}\lambda^{+}(k)}{2L},\;k=1,2,\ldots ,l.$$

## 3. Two-Dimensional Ising Model

Let us discuss the same problem for the two-dimensional Ising model with the periodic boundary conditions. Now the spins are at the nodes of a square lattice of the size L×L, L=2l+1; each spin has l pairs of the neighbors along the horizontal axis, l pairs of the neighbors along the vertical axis, and a lot of neighbors that are outside these axes.

Let $w(k_{1},k_{2})$ be the constant of interaction of the spins spaced by $k_{1}$ steps along the vertical axis and by $k_{2}$ steps along the horizontal axis, $0\leq k_{1},k_{2}\leq l$. Since there is no self-interaction in the system, we set w(0,0)=0. Next, for the sake of simplicity and uniformity we add a unit matrix J(0)=I to the set of the circulant matrices ${\left\{J(k)\right\}}_{k=1}^{l}$ and an L-dimensional vector λ(0)=(1,1,…,1) to the set of the λ-vectors (3). Then, according to [9] in the two-dimensional case, we can present the connection matrix $U_{2}$ as
(7)$$U_{2}=\sum_{k_{1}=0}^{l}\sum_{k_{2}=0}^{l}w(k_{1},k_{2})\cdot J(k_{1})⊗J(k_{2}).$$

Due to the commutativity of the matrices J(k), the eigenvectors of the matrix $U_{2}$ are the Kronecker products of eigenvectors of the one-dimensional model:$$F_{\alpha_{1}\alpha_{2}}=f_{\alpha_{1}}⊗f_{\alpha_{2}},\;\alpha_{1},\alpha_{2}=1,2,\ldots ,L.$$

The equalities $U_{2}F_{\alpha_{1}\alpha_{2}}^{+}=\mu_{2}(\alpha_{1},\alpha_{2})F_{\alpha_{1}\alpha_{2}}^{+}$ define the eigenvalues of the matrix $U_{2}$. We can write them in the form of an $L^{2}$-vector $\mu_{2}$:(8)$$\mu_{2}=\sum_{k_{1}=0}^{l}\sum_{k_{2}=0}^{l}w(k_{1},k_{2})\cdot \Lambda (k_{1},k_{2}),\;\mathrm{where}\;\Lambda (k_{1},k_{2})=\lambda (k_{1})⊗\lambda (k_{2}).$$

Since the vectors λ(k) are mutually orthogonal, any two vectors $\Lambda (k_{1},k_{2})$ that differ by at least one pair of the indices are also orthogonal.

The equality (8) allows us to solve the inverse problem for the two-dimensional Ising model, which is to determine the interaction constants $w(k_{1},k_{2})$, which generate the known spectrum $\mu_{2}$ of the connection matrix (7). It is easy to see that
$$w(k_{1},k_{2})=\frac{\mu_{2}\Lambda^{+}(k_{1},k_{2})}{{\left\|\Lambda (k_{1},k_{2})\right\|}^{2}},\;\mathrm{where}\;k_{1},k_{1}=0,1,\ldots l.$$

Let us note that ${\left\|\Lambda (k_{1},k_{2})\right\|}^{2}={\left\|\lambda (k_{1})\right\|}^{2}\cdot {\left\|\lambda (k_{2})\right\|}^{2}$ and if both the indices $k_{1},k_{2}\neq 0$, then ${\left\|\Lambda (k_{1},k_{2})\right\|}^{2}={\left(2L\right)}^{2}$. However, when one of the indices $k_{i}$ is equal to zero, we need to keep in mind that ${\left\|\lambda (0)\right\|}^{2}=L$.

## 4. *d*-Dimensional Ising Model

### 4.1. Periodic Boundary Conditions

In this case, we can directly generalize the arguments of the previous section to the case of the d-dimensional lattice. Now, the distances $k_{1}$, $k_{2}$…$k_{d}$ between the two spins along all the d axes define the interaction constants $w\;(k_{1}\;,k_{2},\ldots ,k_{d})$. To avoid the self-interaction, we again introduce an additional constant w (0,0,…,0)=0 and make use of the previously introduced unit matrix J(0)=I. Then, the connection matrix for the d-dimensional lattice takes the form [9]:$$U_{d}=\sum_{k_{1}=0}^{l}\sum_{k_{2}=0}^{l}\ldots \sum_{k_{d}=0}^{l}w(k_{1},k_{2},\ldots ,k_{d})\cdot \;J(k_{1})⊗J(k_{2})⊗\ldots ⊗J(k_{d}).$$

Since J(k) are the commute matrices, the eigenvectors of the matrix $U_{d}$ are the Kronecker products of eigenvectors of the one-dimensional model,
$$F_{\alpha_{1}\alpha_{2}\ldots \alpha_{d}}=f_{\alpha_{1}}⊗f_{\alpha_{2}}⊗\ldots ⊗f_{\alpha_{d}},\;\alpha_{i}=1,2,\ldots ,L,\;i=1,2,\ldots ,d,$$
and the corresponding eigenvalues $\mu_{\alpha_{1}\alpha_{2}\ldots \alpha_{d}}$ are equal to
(9)$$\mu_{\alpha_{1}\alpha_{2}\ldots \alpha_{d}}=\sum_{k_{1}=0}^{l}\sum_{k_{2}=0}^{l}\ldots \sum_{k_{d}=0}^{l}w(k_{1},k_{2},\ldots ,k_{d})\prod_{i=1}^{d}\lambda_{\alpha_{i}}(k_{i}),$$
where $\lambda_{\alpha_{i}}(0)=1$, $\alpha_{i}=1,2,\ldots ,L$, and i=1,2,…,d. The same way as in Equation (8), we define $L^{d}$-dimensional vectors $\Lambda (k_{1},\ldots ,k_{d})$ as the Kronecker products of the L-dimensional vectors λ(k):(10)$$\Lambda (k_{1},\ldots ,k_{d})=\lambda (k_{1})⊗\ldots ⊗\lambda (k_{d}).$$Because the vectors λ(k) are mutually orthogonal, any two $L^{d}$-dimensional vectors $\Lambda (k_{1},\ldots ,k_{d})$ and $\Lambda (k_{1}{}^{\prime },\ldots ,k_{d}{}^{\prime })$ that differ by at least one pair of the indices are also orthogonal. Let us introduce an $L^{d}$-dimensional vector $\mu_{d}$, whose coordinates are the values $\mu_{\alpha_{1}\alpha_{2}\ldots \alpha_{d}}$ defined by the equalities (9). Then, we can rewrite Equation (9) in the vector form
$$\mu_{d}=\sum_{k_{1}=0}^{l}\sum_{k_{2}=0}^{l}\ldots \sum_{k_{d}=0}^{l}w(k_{1},k_{2},\ldots ,k_{d})\cdot \Lambda (k_{1},k_{2},\ldots ,k_{d}).$$

The last expression allows us to solve the inverse problem for a hypercubic lattice of an arbitrary dimension d and reconstruct the interaction constants $w(k_{1},k_{2},\ldots ,k_{d})$ when we know the spectrum $\mu_{d}$. The formulas that define these constants are
(11)$$w(k_{1},k_{2},\ldots ,k_{d})=\frac{\mu_{d}\Lambda^{+}(k_{1},k_{2},\ldots ,k_{d})}{{\left\|\Lambda (k_{1},k_{2},\ldots ,k_{d})\right\|}^{2}},\;\mathrm{where}\;k_{1},k_{2},\ldots ,k_{d}=0,1,\ldots l.$$

We can write Equation (11) in another form. Indeed, with the account of the collinearity condition (4) we rewrite Equation (10) as
$$\Lambda (k_{1},\ldots ,k_{d})=2^{D/2}L^{d/2}R_{k_{1}k_{2}\ldots k_{d}},$$
where
$$R_{k_{1}k_{2}\ldots k_{d}}=f(k_{1}+1)⊗\ldots ⊗f(k_{d}+1),\;D=d-\sum_{i=1}^{d}\delta_{0,k_{i}}.$$ Then, Equation (11) takes the form
$$w(k_{1},k_{2},\ldots ,k_{d})=\frac{1}{2^{D/2}L^{d/2}}\mu_{d}R_{k_{1}k_{2}\ldots k_{d}}^{+},\;k_{1},k_{2},\ldots ,k_{d}=0,1,\ldots l.$$

### 4.2. Free Boundary Conditions

In this case, there are L−1 matrices J(k), and they do not commute [8]. This means that we cannot express the eigenvectors of a combination of the matrices J(k) with different k in terms of the eigenvectors of these matrices. Consequently, the approach described above does not fully apply. However, even in this problem we can use its restricted version.

Let us explain this statement. The difficulty of this problem is that even analyzing the one-dimensional Ising system with free boundary conditions, we can account for the interactions with the nearest neighbors only. However, in the two-dimensional case each spin has four nearest neighbors, which belong to the first coordination sphere, as well as four next-nearest neighbors, belonging to the second coordination sphere. In addition, each spin from the second coordination sphere interacts with two spins from the first coordination sphere. Then, only the Kronecker products J(0)⊗J(1) and J(1)⊗J(1) enter Equation (7). These Kronecker products have the same set of the eigenvectors and, consequently, all the arguments of Section 2 remain valid.

In a similar way, we can show that for the three-dimensional model, it is possible to account for interactions with spins from the first three coordination spheres that is with the nearest, the next-nearest, and the next-next-nearest neighbors [8]. In the general case of the d-dimensional system, we can take into account interactions with the spins belonging to the first d-coordination spheres.

## 5. Discussion and Conclusions

In the present paper, we summarize a series of our publications where we published the results of our analysis of the spectral properties of the Ising connection matrix for the hypercubic lattice of the arbitrary dimension d≥1. The key idea is that the d-dimensional connection matrix can be reduced to the Kronecker products of the connection matrices for the one-dimensional Ising model (see [9]).

In the case of the periodic boundary conditions, all the matrices J(k) commute [8,12]. Consequently, all these matrices, their linear combinations $\sum_{k}^{}w(k)J(k)$, or even the products $J(k_{1})\cdot J(k_{2})\cdot \ldots \cdot J(k_{d})$ have the same set of the eigenvectors. This property allows us to include in the consideration the interactions between spins that are arbitrarily far apart.

In the case of the free boundary conditions, the matrices J(k) do not commute [8]. This means that we cannot express the eigenvectors of a linear (or nonlinear) combination of the matrices J(k) in terms of the eigenvectors of these matrices. Consequently, our approach does not fully apply. However, here, we can use its restricted version, when for a d-dimensional lattice we account only for interactions with the spins on the first, second, …, and the d-th coordination spheres (see Section 4.2).

We have checked that our results for periodic boundary conditions coincide with the results that can be obtained by the Fourier transform approach for translationally invariant Hamiltonians [13]. However, we would like to stress that our method of the Kronecker products has a benefit of also being applicable in the case of free boundary conditions, where translationally invariant basis is not helpful. For both these types of boundary conditions, our method yields compact expressions for the eigenstates and eigenvectors in the case of a lattice with arbitrary dimension and with any number of neighbors taken into account. We believe that this is unique for our method and does not have any known analogues.

In the Ising model, the connection matrix J defines the distribution of the energy density $E(s)\sim -sJs^{+}$ of the states $s=(s_{1},\ldots ,s_{N})$, where $s_{i}=\pm 1$. Consequently, the eigenvalues and eigenvectors of the connection matrix define the energy distribution and such macroscopic characteristics of statistical physics as the free energy, the magnetization, the internal energy, and the heat capacity. Nevertheless, up to now it was not evident how the spectral characteristics could help in analyzing the properties of the Ising systems.

However, the authors of a recently published paper [14] successfully applied spectral characteristics of the connection matrix when examining the elementary excitations of the multidimensional systems of interacting spins, such as the Ising models, the Heisenberg models, and the abelian Kitaev anyons. Our results proved to be very helpful for this type of calculation.

## Data Availability

Not applicable.

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
