# Peer review of "Generalized Solution of Inverse Problem for Ising Connection Matrix on d-Dimensional Hypercubic Lattice"

_entropy, 2022, doi:10.3390/e24101424_

Round 1

Reviewer 1 Report

In the present paper the authors survey and generalize the analysis of eigenvalues of connection/interaction matrices for the Ising type models on d-dimensional hypercubic lattices. This work is a continuation of their previous work [2-5] and it is motivated by the success of a previous paper by Dixon et al. [1] and a very recent one [8]. Their most relevant results are related to the comparison of systems with periodic and free boundary conditions. The research subjects fit to the topics of the Special Issue "Ising Model: Recent Developments and Exotic Applications II". In short, I can recommend the publication of this MS after some optional revisions/extensions detailed below.

Detailed critical remarks

1.) The presentation and surveying the relevance of the present subjects can be improved by discussing several systems (detailed in the announcement of the present special issue) where the concept and approaches of Ising models are widely utilized.

2.) The analyses of systems with periodic boundary conditions is restricted to chains (or d-dimensional lattices with linear sizes) that consist of an odd number of spins. Why is it necessary? Furthermore, I think that the discrete Fourier components are eigenvectors for the Hamiltonian of a lattice system exhibiting translation invariance (for periodic boundary conditions). This feature is exploited in many systems of solid state physics. I suggest mentioning these relationships.

3.) The scientific value of this work can be increased by emphasizing the relevance of boundary conditions for biological, social, and economical systems where Ising type models are widely used.

Reviewer 2 Report

I think this manuscript is empty. The authors should to  rewrite the manuscript. As a Math,  more derivation in Equations  of the section 4 should be given with proofs of theorems and so on. 

Reviewer 3 Report

The paper presents generalized solution of inverse problem for Ising connection matrix on d-dimensional hypercubic lattice. Authors construct the connection matrix of the multidimensional model and to write down its eigenvectors and eigenvalues.

The paper is clearly written, the expressions are clear and repeatable.  The paper fails to demonstrate the uniqueness and research contributions of this study. The descriptions and result discussions are not clear as well. 

For this reason the paper need major revisions.

Here are detailed questions and comments for the paper:

1. The introduction does not provide sufficient background, must be extended.

2. The novelty of the paper must be described.

3. Add some applications that confirm your applications.

Best regards

Round 2

Reviewer 2 Report

I think the authors addressed all queries pointed in my previous report and now the paper can be accepted.

Reviewer 3 Report

Dear Authors.

I read your revised work and your answers to my questions, now the work can be published.